# Evaluation of a Structured Light Scanner for 3D Facial Imaging: A Comparative Study with Direct Anthropometry

**DOI:** 10.3390/s24165286

**Published:** 2024-08-15

**Authors:** Martin Major, Bence Mészáros, Tamás Würsching, Melinda Polyák, Gábor Kammerhofer, Zsolt Németh, György Szabó, Krisztián Nagy

**Affiliations:** 1Department of Oro-Maxillofacial Surgery and Stomatology, Semmelweis University, 1085 Budapest, Hungarykammerhofer.gabor@semmelweis.hu (G.K.);; 2Pediatric Center, Semmelweis University, 1085 Budapest, Hungarynagy.krisztian@semmelweis.hu (K.N.); 3Department of Restorative Dentistry and Endodontics, Semmelweis University, 1085 Budapest, Hungary; polyak.melinda9409@gmail.com

**Keywords:** structured light scanner, 3D facial imaging, anthropometry, traditional caliper

## Abstract

This study evaluates the accuracy and repeatability of craniofacial measurements with a 3D light scanner, specifically the EINSTAR scanner, in comparison to traditional caliper measurements for facial anthropometry. Eleven volunteers were assessed by two examiners, one experienced and one inexperienced, who performed direct caliper measurements and indirect measurements using the scanner. Results indicated minimal differences between caliper and scanner results, with overall high accuracy and reliability demonstrated by correlation coefficients. Despite the slightly longer scanning time, the benefits of 3D imaging, including detailed surface mapping and virtual modeling, justify its integration into clinical practice, particularly in maxillofacial surgery and craniofacial assessment. Craniofacial measurements obtained with the EINSTAR scanner showed excellent reliability and accuracy, which qualifies this method for clinical and scientific use.

## 1. Introduction

Three-dimensional (3D) face-scanning technology is employed in a multitude of fields, including security, healthcare, and entertainment. Its applications include the accurate identification of individuals, the improvement of diagnostic capabilities, and the enhancement of user experiences in virtual environments [1,2,3].

Three-dimensional (3D) facial imaging systems are valuable tools with a lot of potential for facial evaluation. They can be used for pre-surgical planning, post-operative assessment of facial symmetry, and surgical shape changes for 3D facial recognition, and also in clinical forensic medicine. The correlation between skeletal and soft-tissue morphology makes them particularly useful [4,5]. Soft tissue facial anthropometry is essential in clinical practice as it provides objective information on craniofacial growth in patients and assists in surgical planning and outcome assessment [6]. These imaging systems also allow for accurate diagnosis of various syndromes by assessing normal and abnormal growth, aiding in maxillofacial surgical planning, and facilitating orthodontic treatment [7]. For the evaluation of facial appearance of patients with a cleft lip, direct clinical anthropometry, 2D and 3D imaging, and clinical videography are used to assess the oral and maxillofacial appearance qualitatively and quantitatively [8,9]. 

Three-dimensional facial recognition technology has emerged as a critical tool for security and identity verification, offering superior accuracy over traditional 2D methods. Three-dimensional facial recognition is also a possible major application for 3D facial scanning. The existing literature contains several studies that examine the historical development and recent advancements in 3D face recognition. These studies focus on pose, expression, and occlusion-invariant techniques [10]. Meghan Li et al. have provided a comprehensive survey of 3D face recognition, covering both traditional and modern methods. They discussed feature extraction, classification, and challenges like pose variations, lighting, and spoofing, along with an in-depth analysis of commonly used 3D face datasets and current trends in the field [11]. Insaf A et al. examined face recognition, noting that 2D methods are accurate in controlled environments but sensitive to lighting and expression changes. They also discussed ongoing advancements in 3D approaches and deep learning to address these challenges [12].

The advancements in 3D imaging can significantly enhance applications ranging from access control to forensic investigations [10,11,12]. In clinical forensic medicine, the accurate assessment of traumatic injuries is essential for legal and medical purposes. Advanced imaging techniques like 3D scanning can improve the assessment of injuries and provide critical information to assist in the investigation and documentation of trauma cases [13]. In forensic pathology, 3D scanning using LiDAR-equipped devices offers a precise and reproducible way to document autopsy findings, surpassing the limitations of traditional photography [14].

Over the past few decades, technological advancements have replaced traditional direct anthropometry, which involves using rulers and calipers, and two-dimensional (2D) photography with non-invasive three-dimensional (3D) surface-imaging methods. The conventional technique involves taking measurements directly on the face using a caliper. This method is simple and non-invasive, and does not require expensive equipment [15]. The 2D system is cost-effective, but facial proportion assessments can be time-consuming and require patient co-operation. Additionally, these assessments may only identify the most obvious disproportions of the face [16].

There are several different techniques for depicting the soft tissues of the face (e.g., ultrasound, laser scanning, and soft-tissue data obtained from CTs), but due to its accuracy, speed, and reliability, digital stereophotogrammetry is the most reliable method [17,18]. Digital stereophotogrammetry is one of the most commonly used techniques for 3D facial-surface imaging today [7,19]. This technique captures images of the facial surface using multiple digital single-lens reflex cameras (digital SLR or DSLR) with overlapping fields of view. The images are then merged using software to create a single 3D model. The facial geometry is represented as a dense point cloud, and the facial skin texture is realistic. Stereophotogrammetry is currently the most promising method of soft-tissue evaluation [20,21,22]. Detailed measurements can be collected on the resulting 3D facial models, such as distances, angles, curvatures, volumes, and surface areas. Three-dimensional images can display the patient from any angle and are quicker to capture than traditional clinical photography, which requires multiple views. In comparison to direct anthropometry, measurements can be taken after the visit without requiring additional time with the patient, and the impact of soft-tissue deformation on variability is reduced [20,23].

In medicine, 3D imaging using 3D digital light scanners is gaining popularity due to increased availability and affordability. This allows for detailed 3D information to be obtained and post-processed in various ways on a personal computer [24]. The digital scanning devices that collect reflected light and images are used to perform digital scanning of the face to analyze the depth information and texturing of 3D objects. Three-dimensional scanning devices have their own advantages and limitations, not only in terms of accuracy but also in terms of data acquisition, processing time, applications, and ease of use [25]. It is a non-invasive method that avoids compressing skin tissue and causing discomfort to the patient [26]. Currently, there are concerns regarding the accuracy of 3D scanners. In order to use 3D data for planning surgical procedures, it is necessary for the scanners to be highly accurate in obtaining and computing 3D data [17]. 

In spite of reports on the reliability of face measurement methods, there is a need to investigate the scanning time, data processing, and reliability of techniques. Technical innovations and the appearance of newer surface scanners make 3D surface-scanning techniques more accessible. However, their clinical applicability must be preceded by validation studies [15,17,18,25,27,28,29,30,31,32].

The objective of this study was to assess the accuracy and reproducibility of a light scanning device that is relatively affordable and straightforward to use, thereby making it accessible to a wide range of users, and also to compare it with manual caliper measurement method. The EINSTAR scanner’s portability allows for convenient on-site data collection, while its user-friendly interface ensures efficient operation by clinical staff. Furthermore, its cost-effectiveness makes it accessible for a wide range of clinical and research applications, without compromising on accuracy or reliability. Studies have highlighted the importance of such devices in improving the accessibility and efficiency of 3D imaging in clinical settings [25,27,33]. The study also aimed to determine the suitability of this method for clinical and scientific applications. The authors intend to pursue further research with the EINSTAR 3D (Shining 3D, Hangzhou, China) scanner in the field of maxillofacial surgery. In their studies, the authors will investigate pre-operative planning and the outcome assessment of orthognathic surgery and cleft lip surgery, as well as post-operative swelling [15,16,34].

## 2. Materials and Methods

### 2.1. Subjects

Eleven volunteers were enrolled for this study, six of them female and the five male. Their age ranged between 18 and 58 years (mean = 31). The sample size was calculated based on a previous study. Basier Ngueira et al. evaluated the accuracy of a new photogrammetric scanner using 11 participants and found that it produced facial measurements with high intra- and inter-examiner reliability (ICC > 0.9) and a minor variation of less than 2.0 mm, except for three measurements. The 3D cloner scanner demonstrated clinically acceptable accuracy, comparable to a digital caliper, with a mean difference of −0.8 ± 1.2 mm [31]. Exclusion criteria were history of either unilateral or bilateral facial procedures, surgery, trauma, or congenital conditions that could alter facial anatomy [31].

Written informed consent was obtained from all participants for the use of their facial images and personal data for research purposes and publication before inclusion in the study.

The study was approved by the Ethics in Research Committee of Semmelweis University (SE RKEB: 265/2019)

### 2.2. Examiners

Two examiners were involved in the study: the first (TW) had prior experience with the scanner and cephalometric analysis, and the second was untrained (MM). Facial landmarks were placed by the two examiners together, and afterwards both of them took direct measurements with the caliper and then did facial scanning and indirect measurements on the facial scan of each patient.

### 2.3. 3D Imaging and Measurements

The subjects were asked to remove any jewelry and hair from the face and forehead area to give better exposure to the area of interest. Male volunteers were asked to shave off any facial hair.

A total of 14 cephalometric landmarks were defined using the mirror application (Canfield Scientific; Parsippany, NJ, USA) [27] and a total of 11 linear distances were measured. The reference points were marked prior to measurement with a Devon skin marker (Covidien Inc., Minato City, Japan) as this had been proven to increase accuracy. These are important cephalometric points, and the aim was to measure horizontal, vertical, and oblique dimensions to see if there was any directional distortion. In analogous studies, the selected points, angles, and distances were also selected at the discretion of the investigator. The objective of the authors was to select points that had already been chosen in other studies, but did not copy one set of points from another article [15].

The following reference points were chosen (Figure 1):G—Glabella: the most anterior midpoint on the fronto-orbital soft tissue contour;N—Nasion: the midpoint on the soft tissue contour of the base of the nasal root, at the level of the frontonasal suture;Sn—Subnasale: the midpoint on the nasolabial soft tissue contour between the columella crest and the upper lip;Prn—Pronasale: the most anterior midpoint of the nasal tip;Al—Alare right (Alr) and left (All) is the most lateral point on each alar contour;Pg—Pogonion is the most anterior midpoint of the chin;Me—Menton is the most inferior midpoint on the soft tissue contour of the chin;Ch—Chelion right (Chr) and left (Chl) is the point located at each labial commissure;Ex—Exocanthion right (Exr) and left (Exl) is the soft tissue point located at the outer commissure of each eye fissure;Go—Gonion right (Gor) and left (Gol) is the most lateral point on the soft tissue contour of each mandibular angle.

The following linear measurements were obtained from the reference points:Glabella—Subnasale;Subnasale—Menton;Nasion—Pronasale;Subnasale—Pronasale;Exocanthion Right—Exocanthion Left;Cheilion Right—Cheilion Left;Exocanthion Right—Cheilion Right;Exocanthion Left—Cheilion Left;Gonion Right—Pogonion;Gonion Left—Pogonion;Alare Right—Alare Left.

During measuring, the participants were asked to remain seated, maintaining a natural head position with relaxed facial muscles, eyes closed, and without facial expression. A digital caliper (Silverline, Iddesleigh, UK) with 0.01 mm accuracy was used for direct measurements on the subjects’ faces. To avoid distortion, the tip of the caliper was lightly positioned on the marked points with no pressure on the soft tissue during measurement. For facial scanning, the EINSTAR scanner was used (Shining 3D, Hangzhou, China) (Figure 2). Table 1 shows the technical specifications of the scanner. The following settings were used for scanning:Scan mode: portrait scan;Alignment: hybrid;Resolution: 0.2 mm;Data Quality Indicator: on;Hair more: off.

Scanning took place under natural ambient light. Post-processing of the scanned data and indirect measurements on the facial scans were all done using the scanner’s proprietary ExStar software (v1.2) provided by the manufacturer. During scanning, the scanner needs to be connected to a PC using MS Windows. The recommended hardware settings are: OS: Win 10/11, 64 bit; graphics card: NVIDIA GTX1060; video memory: ≥6 GB; processor: I7-11800H; memory: ≥32 GB (Figure 3).

### 2.4. Time Analysis

Time was measured using a stopwatch (ACCUSPLIT Pro Survivor—A601X Stopwatch, Torrance, CA, USA). For the direct measurements with the caliper-only one, for the measurement with the 3D scanner three separate time periods were registered: one for scanning, one for post-processing of the data, and one for the indirect measurement on the facial scan.

### 2.5. Statistical Analysis

Statistical analysis was conducted using IBM SPSS version 25. The *Kolmogorov–Smirnov* test was used to assess normality. A 95% confidence interval and a significance level of *p* < 0.05 were defined. Means were compared using a *t*-test, and correlation was assessed using *Spearman’s correlation*.

As the sample distribution did not deviate from normal, the *t-test* was deemed appropriate for comparing the data.

To compare scanning, time-descriptive statistics were used. The following flowchart illustrates the structure of the research project (Figure 4).

## 3. Results

The statistical analysis was conducted using IBM SPSS version 25. A Kolmogorov–Smirnov test for normality was employed. A 95% confidence interval and a significance level of *p* < 0.05 were defined a priori. A *t*-test was employed for the purpose of comparing means, while a Spearman correlation was utilised.

The distribution of the sample did not deviate from the normal distribution, thus rendering a *t*-test an appropriate statistical tool for comparing the data.

The two-sample *t*-test requires that both samples have a normal distribution and that the standard deviation squares are equal. This was fulfilled in our present sample, with the distribution of our small sample from the population being normal. The applicable test function is as follows:$$t=\frac{\bar{X_{1}}-\bar{X_{2}}}{S_{p}\sqrt{\frac{1}{n_{1}}+}\frac{1}{n_{2}}}$$

Degree of freedom: *n*_1_ + *n*_2_ − 2

Common scattering square formula:$$s_{p}^{2}=\frac{\left(n_{1}-1\right)s^{2}\frac{1}{1^{-}}\left(n_{2}-1\right)s_{2}^{2}}{n_{1}+n_{2}-1}$$

The data from 11 subjects were analyzed. Table 2 shows the mean caliper measurements of two examiners and the difference between them.

The measurements taken by the two examiners were not significantly different, indicating similar accuracy. The only notable difference was at the subnasale–menton point, where the second examiner recorded a smaller distance (highlighted in red as significant).

The correlation test results show a significant correlation between the caliper averages of Examiner 1 and Examiner 2 for all measurements, indicating measurement accuracy. The correlation coefficient value indicates an excellent correlation between the variables. Regarding the subnasale–pronasale relationship, the coefficient indicates a moderate result, confirming the accuracy of the measurement, but it is weaker than in other cases. Table 3 shows the mean scanner measurements of two examiners and the difference between them.

The results show that there is no significant difference between the measurements of the two inspectors at any point. The correlation analysis shows that each measurement is closely related according to the correlation coefficient, thus justifying the identity of the results of the two examiners.

The correlation analysis shows that the average of each measurement has a close relationship according to the correlation coefficient value, so the scanner results of the two examiners can be considered identical. Table 4 shows the measurements done by Examiner 1 with caliper and scanner, and the difference between them.

There was no significant difference between the average measurements of the first examiner’s caliper and scanner measurements, indicating that both measurements reflect the same result.

The correlation coefficients completely cover the accuracy of the measurements, but there is only moderate agreement in the case of the subnasale-pronasale distance, while there was no significant result in the chelion–chelion averages, thus not confirming the accuracy of the two measurements. Table 5 shows the measurements done by Examiner 2 with caliper and scanner, and the difference between them.

In the case of Examiner 2, significant differences were observed between the caliper and scanner measurements at two points (subnasale–menton and exocanthion R–cheilion R), which may be due to the examiner’s inexperience or the difficulty of the measurement points during manual measurement. The caliper measurements of Examiner 2 were typically smaller than those measured by the scanner. In the case of Examiner 1, smaller caliper results were also typical, but with smaller differences than in the case of Examiner 2.

By examining the correlation coefficients, the accuracy of the measurements can be confirmed. However, when compared to the averages of the chelion–chelion measurements, the relationship appears weak, and the accuracy of the measurement cannot be confirmed. A relationship between the caliper and scanner measurements has also been confirmed through correlation analysis, indicating an overall correlation between the two.

When comparing measurements made using the caliper and the scanner, a mean deviation of −0.36 mm ± 1.52 mm (range: −3.79 to 4.9 mm) was observed.

Table 6 summarizes the time required for the measurements. The direct technique resulted in an average measurement time of 02:25 for an experienced investigator and 2:53 for an inexperienced investigator. The measurement time with the 3D scanner was divided into three parts: scanning time, image processing, and measurement on mesh. The experienced examiner was faster in each of these parts. The experienced investigator and the inexperienced investigator both had shorter actual scanning times (01:18 and 01:40, respectively) compared to direct measurements with the caliper.

A comparison of measurements taken with calipers and those obtained through 3D scanning reveals a number of notable differences. Calipers are a cost-effective option for direct measurements, however, they are time-consuming and necessitate the patient’s presence. Furthermore, they may also cause tissue distortion, which can lead to errors in measurement. In contrast, 3D scanning, though more expensive, offers a faster process through indirect virtual modelling, with post-processing required. It allows for repeated measurements on a 3D model, reducing the need for a patient to be present, and eliminating tissue distortion issues. Additional benefits of 3D scanning include data storage and exportation, the ability to measure arbitrary points subsequently, and the accurate mapping of soft tissue and complex morphologies. Table 7 points out the limitations and benefits of both approaches.

## 4. Discussion

In this clinical research, we employed the EINSTAR 3D light scanner due to its mobility, ease-of-use, and affordability, making it a superior choice compared to other scanners in the field [25,27,33].

In maxillofacial surgery, the possibilities provided by 3D scanners are used in many clinical processes (diagnosis, planning, and follow-up) [35,36,37]. Two of the most studied photogrammetric devices are 3DMD [30,38,39,40] and Vectra [6,41], which are considered the gold standards. Maal. T. J., et al. evaluated registration procedures for 3D facial photographs in oral and maxillofacial surgery, finding surface-based registration, particularly with pre-processed images, to be highly accurate. Both Maxilim and 3dMD software v3.0 showed similar precision, though accuracy decreased slightly with changes in facial expression [38]. Camison L. et al. validated the hand-held Vectra H1 3D surface-imaging system against the 3dMDface system, finding that it produced highly comparable measurements with an average technical error of 0.84 mm, and an RMS error of 0.43 mm. These results suggest that the Vectra H1 system is sufficiently accurate for most clinical applications [6].

Craniofacial anthropometry is a method that is widely used in dentistry and maxillofacial surgery [42], with applicability in the study of facial growth. Zhang S. et al. used 3D white-light scanning to study the growth of nasolabial units in 528 Chinese children, finding that these units grow most rapidly from the ages of 0 to 2, with similar growth patterns across sexes [43]. Facial appearance can be a significant clue in the initial identification of genetic conditions [44]. Hammond P. et al. described the potential of computer-based models of three-dimensional (3D) facial morphology to assist in dysmorphology training, in clinical diagnosis, and in multidisciplinary studies of phenotype–genotype correlations [45]. In addition, the literature indicates that three-dimensional imaging is a valid method for planning and evaluating the results of orthodontic and orthognathic surgical treatment. Hayeer MY et al. validated a new facial volumetric assessment method using stereophotogrammetric models and a software-based facial analysis tool, finding the “tetrahedron formation” method to be the most accurate, with minimal error, especially when a standardized facial expression is maintained during image acquisition [46].

The purpose of this study was to validate the EINSTAR 3D scanner (Shining 3D, Hangzhou, China) in relation to scanning the face for further investigations and researche in the field of maxillofacial surgery, pre-operative planning and outcome assessment of orthognathic surgery, and cleft lip surgery, as well as post-operative swelling and for clinical application, as it was further explained [8,34,46,47].

For this purpose, the conventional caliper measurement method was compared to indirect measurements on a facial scan to determine the distances between pre-defined anthropometric points on the face. Previous studies have shown that facial landmarks can improve the accuracy of anthropometric measurements [15,19]. Franco de Sa Gomes, C., et al. evaluated craniofacial measurements using an Artec Eva 3D light scanner on 15 volunteers, finding that marking facial reference points improved accuracy, achieving around 1 mm accuracy with marks and 2 mm without. While the scanner showed excellent reliability (ICC = 0.92–0.97), it required twice as much time as direct measurement with a caliper [15].

When comparing measurements made using the caliper and the scanner, a mean deviation of −0.36 mm ± 1.52 mm (range: −3.79 to 4.9 mm) was observed. Weinberg et al. conducted a study in which they performed and compared 3D anthropometric measurements on mannequin heads. The study found that differences in measurements on static models were consistently less than 1 mm [39].

In a validation study, volunteers were used to investigate the clinical use of 3D scanners. The study compared measurements from two 3D scanners (Artec and Sense) with measurements from a reference (Vectra) scanner. The mean deviation for the Artec scanner was 0.13 ± 2.71 mm (range: −10.26 to 13.93 mm), and for the Sense scanner it was −0.19 ± 3.81 mm (range: −14.09 to 18.92 mm) [27].

The results of the two examiners, one experienced and one inexperienced, were compared using both a caliper and a 3D scanner. The inter-examiner analysis revealed a significant difference in the subnasale–menton distance in one case when caliper measurements were taken. This difference can be attributed to the varying pressure exerted by the investigators on the moving soft tissue points with the caliper [15]. In all other respects, the measurement results of the two testers are considered identical, whether using the caliper or the scanner.

When analyzing the measurements of each investigator using different methods, distances measured directly were typically shorter than the same distances measured indirectly. This finding is consistent with previous studies [15,48]. However, significant differences were only observed in two cases for the inexperienced examiner (right exocanthion–right cheilion and subnasale–menton). The correlation coefficients for the distance between cheilion–cheilion points indicate a weak relationship (ICC: 0.32) for the experienced examiner, whereas the other examiner demonstrates excellent but moderately good results (ICC: 0.8). When examining the results, it is important to consider that testing on human subjects may result in increased variability and measurement errors due to subtle changes in facial expressions and more complex surface features [9,22]. This effect is particularly noticeable in the perioral region [49].

Similar to a previous study [27], the scanning process took longer (approximately 6 min) compared to the direct measurement technique. However, it is important to note that the time spent on face scanning was actually shorter than that of caliper measurement, even for inexperienced investigators.

Although the 3D scanning method of data collection takes longer overall, surface scanning has several advantages in the context of facial surgery. This technique enables the patient’s face to be modelled with the highest possible accuracy, making it ideal for mapping soft tissue and areas with complex morphologies such as ears, nose, and lips. Additionally, it allows for measurements to be taken in 3D, providing almost unlimited possibilities for measurements on virtual models. It allows for data storage and exportation, repeated measurements on the 3D model, and the creation of databases for longitudinal studies [35].

Although face scanning has advantages, it is important to recognize potential limitations and challenges associated with its implementation. One limitation is the initial cost of acquiring the scanning equipment, which may pose a barrier to adoption for some healthcare facilities. Furthermore, while the accuracy of face scanning technology is generally high, factors such as motion artifacts or patient discomfort during scanning may affect data quality. Further research and technological advancements in the field of medical imaging may be necessary to address these challenges [50].

## 5. Conclusions

In conclusion, this study demonstrates that the EINSTAR 3D light scanner provides highly accurate and reliable craniofacial measurements, comparable to traditional caliper methods. Examiners achieved no significant differences between caliper and scanner results, despite the minor differences observed, especially for less experienced examiners. Despite the slightly longer scanning time, the advantages of 3D imaging, including detailed surface mapping and virtual modeling, support its clinical integration, especially in maxillofacial surgery and craniofacial assessment. The EINSTAR scanner’s excellent performance qualifies it as a viable tool for clinical and scientific use in facial anthropometry. The authors are planning further research in the field of maxillofacial surgery to investigate post-operative oedema, post-operative follow-up of orthognathic surgery, and surgical management of cleft lip patients using the EINSTAR 3D light scanner.

## Figures and Tables

**Figure 1 sensors-24-05286-f001:**
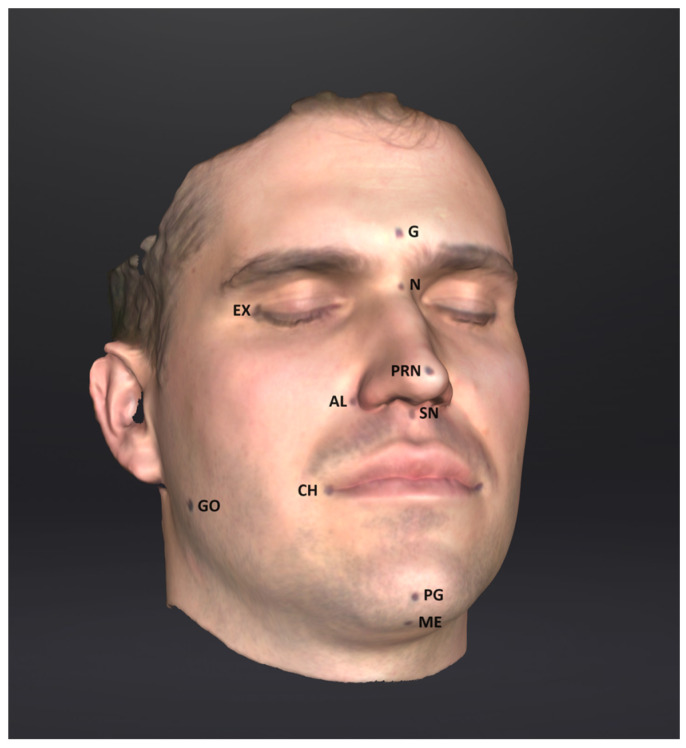
The cephalometric reference points.

**Figure 2 sensors-24-05286-f002:**
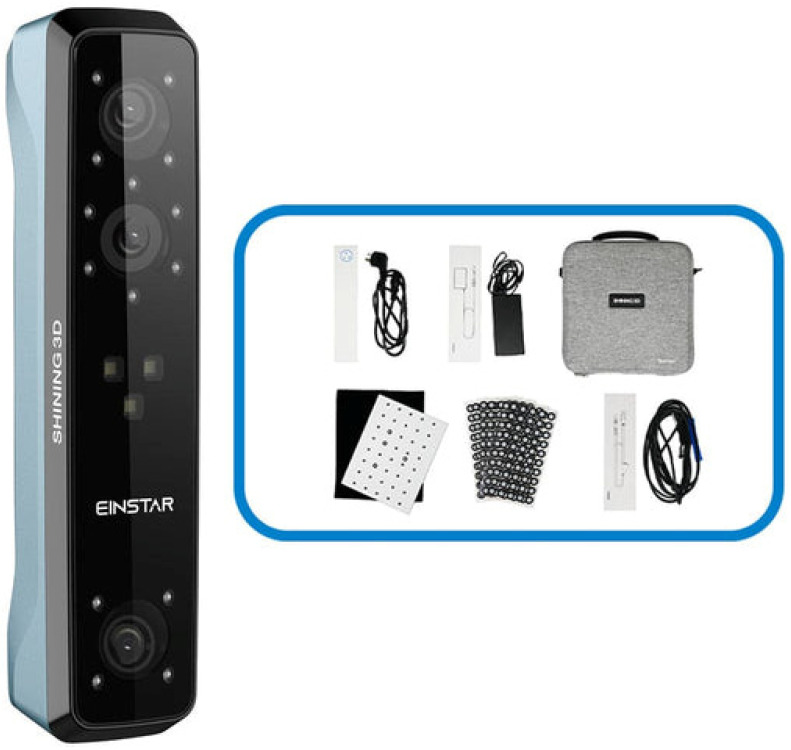
The EINSTAR scanner and accessories (source: https://www.einstar.com/products/prosumer-portable-3d-scanner?variant=43551575507118%20%2Fedition%2F20240814.html) (accessed on 6 August 2024).

**Figure 3 sensors-24-05286-f003:**
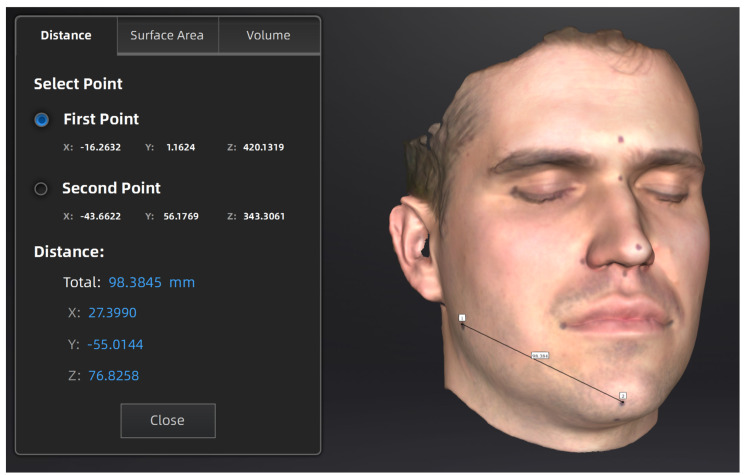
Obtaining linear measurements on the digital model of the face after scanning and post-processing.

**Figure 4 sensors-24-05286-f004:**
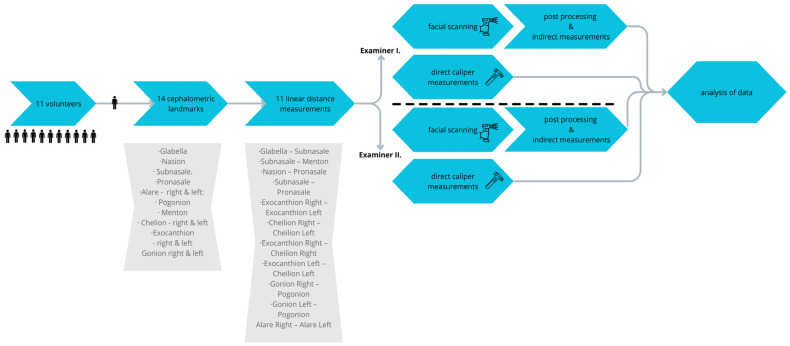
Steps and processes involved in the research.

**Table 1 sensors-24-05286-t001:** Technical specifications of the EINSTAR scanner.

Scan Mode	Structured Light Scan
Point Distance	0.1 mm~3 mm
Light Source	Infrared VCSEL Structured Light
Working Distance	Effective Working Distance: 160 mm–1400 mm; Optimal Working Distance: 400 mm
Depth of Field	160 mm–1400 mm
Maximum Field of View	434 mm × 379 mm (under optimal work distance)
Scan Speed	980,000 points/s, up to 14 FPS
Align Modes	Feature Alignment, Hybrid Alignment, Texture Alignment, GloLeft Markers
Safety	Eye-safe
Texture Scan	Yes
Outdoor Scanning	Yes
Interface	USB2.0 or above
Output Formats	OBJ; STL; PLY; P3; 3MF
Scanner Size	220 mm × 46 mm × 55 mm
Carrying Case Size	245 mm × 245 mm × 90 mm
Scanner Body Weight	500 g
Operating Temperature Range	0–40 °C
Operating Humidity Range	10–90%
Certification	CE, FCC, ROHS, WEEE, KC
Recommended Configuration	OS: Win 10/11, 64 bit; graphics card: NVIDIA GTX1060 (Nvidia, Santa Clara, CA, USA); video memory: ≥6 GB; processor: I7-11800H; memory: ≥32 GB
Basic Computer Configuration	OS: Win 10, 64 bit; graphics card: NVIDIA GTX1050 (Nvidia, Santa Clara, CA, USA); video memory: ≥4 GB; processor: I7-7700H; memory: ≥16 GB

**Table 2 sensors-24-05286-t002:** The caliper measurements of two examiners and the difference between them.

	Examiner 1 (Caliper)	Examiner 2 (Caliper)	Difference of the Means	*t*-Test	Correlation Coefficient (ICC)
Glabella—Subnasale	66.73	66.91	−0.18	0.751	0.954 ***
Subnasale—Menton	69.94	68.53	1.41	0.001	0.990 ***
Nasion—Pronasale	45.16	45.25	−0.09	0.842	0.949 ***
Subnasale—Pronasale	20.98	21.32	−0.34	0.475	0.880 **
Exocanthion—Exocanthion	95.66	96.18	−0.52	0.263	0.990 ***
Cheilion—Cheilion	53.38	53.28	0.1	0.796	0.967 ***
Exocanthion Right—Cheilion Right	71.31	71.04	0.27	0.603	0.963 ***
Exocanthion Left—Cheilion Left	71.08	71.09	−0.01	0.984	0.960 ***
Gonion Left—Pogonion	86.01	85.44	0.57	0.255	0.992 ***
Gonion Right—Pogonion	85.00	84.34	0.66	0.128	0.993 ***
Alare-Alare	34.69	34.21	0.48	0.317	0.933 ***

** *p* < 0.001 *** *p* < 0.0001. Reliability: ICC < 0.4 (poor); 0.4 ≤ ICC < 0.75 (medium to good); ICC ≥ 0.75 (excellent).

**Table 3 sensors-24-05286-t003:** Mean scanner measurements.

	Examiner 1 (Scanner)	Examiner 2 (Scanner)	Difference of the Means	*t*-Test	Correlation Coefficient (ICC)
Glabella—Subnasale	67.41	67.54	−0.13	0.798	0.933 ***
Subnasale—Menton	70.45	70.05	0.4	0.192	0.992 ***
Nasion—Pronasale	44.82	44.65	0.17	0.713	0.948 ***
Subnasale—Pronasale	21.80	21.94	−0.14	0.674	0.922 ***
Exocanthion—Exocanthion	95.06	95.01	0.05	0.048	0.988 ***
Cheilion—Cheilion	53.06	53.13	−0.07	0.948	0.791 ***
Exocanthion Right—Cheilion Right	72.23	72.26	−0.03	0.924	0.986 ***
Exocanthion Left—Cheilion Left	71.26	70.69	0.57	0.177	0.974 ***
Gonion Left—Pogonion	86.98	86.31	0.67	0.080	0.994 ***
Gonion Right—Pogonion	84.93	84.71	0.22	0.522	0.995 ***
Alare-Alare	34.51	34.68	−0.17	0.682	0.946 ***

*p* < 0.001 *** *p* < 0.0001. Reliability: ICC < 0.4 (poor); 0.4 ≤ ICC < 0.75 (medium to good); ICC ≥ 0.75 (excellent).

**Table 4 sensors-24-05286-t004:** Comparison of the results measured by Examiner 1.

	Examiner 1 (Caliper)	Examiner 1 (Scanner)	Difference of the Means	*t*-Test	Correlation Coefficient (ICC)
Glabella—Subnasale	66.73	67.41	−0.68	0.050	0.978 ***
Subnasale—Menton	69.94	70.45	−0.51	0.193	0.989 ***
Nasion—Pronasale	45.16	44.82	0.34	0.360	0.970 ***
Subnasale—Pronasale	20.98	21.80	−0.82	0.124	0.860 **
Exocanthion—Exocanthion	95.66	95.06	0.6	0.346	0.982 ***
Cheilion—Cheilion	53.38	53.06	0.32	0.341	0.320 ***
Exocanthion Right—Cheilion Right	71.31	72.23	−0.92	0.048	0.970 ***
Exocanthion Left—Cheilion Left	71.08	72.23	−1.15	0.703	0.965 ***
Gonion Left—Pogonion	86.01	86.98	−0.97	0.063	0.990 ***
Gonion Right—Pogonion	85.00	84.93	0.07	0.834	0.998 ***
Alare-Alare	34.69	34.51	0.18	0.749	0.904 ***

** *p* < 0.001 *** *p* < 0.0001. Reliability: ICC < 0.4 (poor); 0.4 ≤ ICC < 0.75 (medium to good); ICC ≥ 0.75 (excellent).

**Table 5 sensors-24-05286-t005:** Comparison of the results measured by Examiner 2.

	Examiner 2 (Caliper)	Examiner 2 (Scanner)	Difference of the Means	*t*-Test	Correlation Coefficient (ICC)
Glabella—Subnasale	66.91	67.54	−0.63	0.270	0.937 ***
Subnasale—Menton	68.53	70.05	−1.52	0.005	0.983 ***
Nasion—Pronasale	45.25	44.65	0.6	0.289	0.928 ***
Subnasale—Pronasale	21.32	21.94	−0.62	0.053	0.935 ***
Exocanthion—Exocanthion	96.18	95.01	1.17	0.685	0.991 ***
Cheilion—Cheilion	53.28	53.13	0.15	0.878	0.800 *
Exocanthion Right—Cheilion Right	71.04	72.26	−1.22	0.009	0.978 ***
Exocanthion Left—Cheilion Left	71.09	70.69	0.4	0.510	0.943 ***
Gonion Left—Pogonion	85.44	86.31	−0.87	0.116	0.989 ***
Gonion Right—Pogonion	84.34	84.71	−0.37	0.426	0.991 ***
Alare-Alare	34.21	34.68	−0.47	0.281	0.928 ***

*p* < 0.05 * *p* < 0.001 *** *p* < 0.0001. Reliability: ICC < 0.4 (poor); 0.4 ≤ ICC < 0.75 (medium to good); ICC ≥ 0.75 (excellent).

**Table 6 sensors-24-05286-t006:** Comparison of the measurements times.

Examiner 1 (caliper)	02:25	Examiner 2 (caliper)	02:53
Examiner 1 (scanner) sum	08:04	Examiner 2 (scanner) sum	08:58
Scanning time	01:18	Scanning time	01:40
Image processing	05:14	Image processing	05:35
Measurement on mesh	01:30	Measurement on mesh	01:42

**Table 7 sensors-24-05286-t007:** Comparison of the measurements.

	Measurement with Caliper	3D Scanning
Price of the device	Inexpensive	Higher-priced
Method of measurement	Direct	Indirect—on a virtual model
Time of the measurement	Time-consuming	Relatively fast scanning Post-processing Indirect measurement
Reproducibility	Presence of the patient required	Repeated measurements on a 3D model are possible
Possible causes of measurement error	Tissue distortion during measurement	Motion artifacts
Other benefits	-	Data storage and exportation Subsequent measurement of arbitrary points Accurate mapping of soft tissue and areas with complex morphology

## Data Availability

Data are unavailable due to privacy or ethical restrictions.

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
