# Peer review of "Evaluation of a Structured Light Scanner for 3D Facial Imaging: A Comparative Study with Direct Anthropometry"

_sensors, 2024, doi:10.3390/s24165286_

Round 1

Reviewer 1 Report (Previous Reviewer 2)

Comments and Suggestions for Authors

The authors attended the reviewer's requirements, the the paper can be accepted in its actual form.

Author Response

Please see attachement.

Reviewer 2 Report (Previous Reviewer 3)

Comments and Suggestions for Authors

The manuscript has been significantly improved and can be published as is, subject to minor corrections, notably in reference 9, where the name of co-author A. O. should be written explicitly.

Comments on the Quality of English Language

The English is correct and can be understood without any problems.

Author Response

Please see attachement.

Reviewer 3 Report (New Reviewer)

Comments and Suggestions for Authors

The authors presented the utilization of a new scanner for 3D facial imaging. The study does not provide any new methods but tries to demonstrate that the utilization of the portable scanner is suitable for clinical practice. Because of that, I am missing a more in-depth comparison of the tested methods. Why did not the authors use also some other scanners? Since it is a comparative study, I would appreciate a table pointing out the limitations and benefits of both approaches (it is partly written in the text). Furthermore, if all of the points are selected before utilizing the approaches, how long did this process take for the examiners? How would the results differ if the points had not been selected prior to the utilization of the scanner? Some parts in the discussion section are also more suitable for the introduction. 

Comments on the Quality of English Language

Minor corrections

Author Response

Reviewer 4 Report (New Reviewer)

Comments and Suggestions for Authors

This paper presents a study that evaluates the accuracy and repeatability of craniofacial measurements using a 3D light scanner (EINSTAR) compared to traditional caliper measurements. The researchers have taken a comprehensive approach by comparing the two measurement techniques, which is crucial for validating the reliability and clinical applicability of the 3D scanning method. Involving both experienced and inexperienced examiners in the measurements helps to assess the generalizability of the results and the ease of use of the 3D scanning technology. The 3D scanning results show minimal differences and high correlation coefficients compared to the caliper measurements, suggesting excellent reliability and accuracy of the EINSTAR scanner. The abstract highlights the potential benefits of 3D imaging, such as detailed surface mapping and virtual modeling, which can justify the integration of this technology into clinical practice, particularly in maxillofacial surgery and craniofacial assessment.

There are several limitations that the authors need to address:

·       The authors should provide a clear flowchart or visual representation of the methodology used in the study. A well-designed flowchart would help readers better understand the various steps and processes involved in the research, improving the clarity and transparency of the study.

·       The study lacks mathematical expressions or formulas to clearly articulate the methodology used. Incorporating relevant mathematical expressions would help readers better understand and evaluate the technical aspects of the research approach.

·       The references used are quite old, with 36 out of 41 references being more than five years old. While older references can provide valuable historical context, it is important to also include more recent literature to ensure the review reflects the current state of knowledge in the field.

Author Response

Please see attachement.

Reviewer 5 Report (New Reviewer)

Comments and Suggestions for Authors

This manuscript evaluates the accuracy and repeatability of craniofacial measurements with a 3D light scanner, specifically the EINSTAR scanner, in comparison to traditional caliper measurements for facial anthropometry. This study is timely and important for clinical and scientific use, particularly in maxillofacial surgery and craniofacial assessment. The manuscript is well-written to the knowledge and research area. The core idea seems interesting, but the paper should be improved in some regards. 

1.     Materials and methods - There seems to be a shortage of 11 volunteers, and there is a problem of insufficient data volume when conducting statistical analysis. If possible, the author needs to increase the number of volunteers.

2.     Discussion - The writing is quite cumbersome, and many contents can be introduced in the introduction section. Because some conclusions were not obtained through the author's experimental data analysis.

3.     Reference- The references are generally obsolete, replacing or adding to the relevant research results of the last three years.

4.     Future work can be added to the conclusion.

Comments on the Quality of English Language

Please carefully check grammar, spelling, and other issues.

Author Response

Please see the attechement.

Round 2

Reviewer 5 Report (New Reviewer)

Comments and Suggestions for Authors

This manuscript have solved most of my concerns and now is appropriate for publication in Sensors.

This manuscript is a resubmission of an earlier submission. The following is a list of the peer review reports and author responses from that submission.

Round 1

Reviewer 1 Report

Comments and Suggestions for Authors

This paper presents an accuracy test report for a commerical 3D scanner, technical contribution of the paper is quite limited.

Comments on the Quality of English Language

NA

Reviewer 2 Report

Comments and Suggestions for Authors

This paper presents an evaluation of the accuracy and repeatability of measurements with the EINSTAR 3D scanner.  Before taking a decision about the acceptation or rejection recommendation of this paper, several issues must be attended.

1.        The contribution of this paper is not clear.

2.        The relevance of the obtained results must be clearly explained.

3.        The paper does not present information about previous studied about this topic.  Please explain.

4.        The present studies were carried out with only 11 persons.  This number appears to be small, please explain why this number of persons was used.

5.        Please explain the potential applications of this results.

6.        Why the EINSTAR 3D scanner was chosen.    

7.        Please explain how the points shown in Fig. 1 was chosen.

8.        How the results reported in this paper can be used.  Please explain.

Reviewer 3 Report

Comments and Suggestions for Authors

 The proposed work is interesting, but weren't the results predictable?

Furthermore, the authors proposed maxillofacial surgery and evaluation of the skull and face as possible applications, which is very restrictive. The authors should enrich their manuscript by mentioning other major applications and citing the key references that attach to them, for example in 3D facial recognition [1, 2, 3], and traumatic injuries in clinical forensic medicine [4] and in Forensic Pathology [5]

[1] https://doi.org/10.1016/j.engappai.2022.104669

[2] https://doi.org/10.3390/electronics9081188

[3] https://doi.org/10.1186/s13673-018-0157-2

[4] https://doi.org/10.1016/j.jflm.2018.05.004

[5] https://doi.org/10.3390/bios12020132

Comments on the Quality of English Language

In general, the English used is correct